Genome-wide identification and expression analysis of the Trihelix transcription factor family in potato (Solanum tuberosum L.) during development

Mei Chao 1 2
Liu Yuwei 3 liuyw@hebau.edu.cn
Song Huiyang 1
Li Jinghao 1
Song Qianna 1
Duan Yonghong 1 2
Feng Ruiyun 1 2 Fengruiyun1970@163.com
1 College of Agriculture, Shanxi Agricultural University , Taiyuan , China
2 Key Laboratory of Potato Genetic Improvement and Germplasm Innovation in Shanxi Province , Datong , China
3 College of Life Sciences, Hebei Agricultural University , Baoding , China
Uversky Vladimir
Electronic publication date: 2024 Nov 29
Publication date: 2024
Volume: 12
Electronic Location ID: e18578
Received 2024 May 20; Accepted 2024 Nov 4
Copyright: © 2024 Mei et al.
Copyright year: 2024
Copyright holder: Mei et al.
License: This is an open access article distributed under the terms of the Creative Commons Attribution License, which permits unrestricted use, distribution, reproduction and adaptation in any medium and for any purpose provided that it is properly attributed. For attribution, the original author(s), title, publication source (PeerJ) and either DOI or URL of the article must be cited.
License URL: https://creativecommons.org/licenses/by/4.0/

Keywords: Solanum tuberosum, Trihelix transcription factors, Development, Expression, Network

Funding: Key Research and Development Plan Project of Shanxi Province 202102140601004 Breeding Engineering Project of College of Agriculture, Shanxi Agricultural University YZ2021-04 Doctor Foundation of Shanxi Agricultural University ZB1102 Doctor Foundation of Institute of Crop Science, Shanxi Academy of Agricultural Sciences ZB1901 This work was supported by the Key Research and Development Plan Project of Shanxi Province (202102140601004), the Breeding Engineering Project of College of Agriculture, Shanxi Agricultural University (YZ2021-04), the Doctor Foundation of Shanxi Agricultural University (ZB1102)), and the Doctor Foundation of Institute of Crop Science, Shanxi Academy of Agricultural Sciences (ZB1901). The funders had no role in study design, data collection and analysis, decision to publish, or preparation of the manuscript.

==============================
Trihelix transcription factors (TF) are photoresponsive proteins featuring a characteristic three-helix structure (helix-loop-helix-loop-helix) and contain the Myb/SANT-LIKE (MSL) domain. They perform crucial functions in the development and stress-response of plants. However, the function of the Trihelix TF in potato (Solanum tuberosum L.) remains unknown. In the present study, forty-three StMSLs were characterized in the potato genome and named StMSL1 to StMSL43. Structural domain analysis revealed that motifs 1 and 2 may play a central role in the implementation of trihelix gene functions, and motifs 4 and 9 may be related to specific functions of StMSL. Phylogenetic analysis divided the StMSLs into six groups (SIP1, GT1, GT2, GTγ, SH4 and GT3). The GT3 group, which is rarely identified in the Trihelix TF family, contained three StMSLs. The 43 StMSLs were unevenly distributed on 12 chromosomes in potato and comprised two pairs of tandem duplication and five pairs of segmental duplication genes. Additionally, RNA-Seq analysis found that 36 out of the 43 StMSLs were expressed in at least one of the 12 tissues, with some exhibiting tissue-specific expression patterns. Trihelix transcriptional regulation network analysis identified 387 genes as potential targets of the 36 StMSL genes, and these genes have a wide variety of functions. Furthermore, RNA-Seq analysis revealed that at least 18 StMSLs were upregulated in response to osmotic stress. The induced pattern of eight StMSLs was subsequently validated using qRT-PCR. This study provides a detailed insight into the StMSLs of the potato and lays the foundation for further analysis of the functions of the Trihelix gene in plant development.

Introduction

Transcription factors (TF) are transcriptional regulatory elements that are ubiquitous in eukaryotes (Cooper, 2000). During the evolution of plants, a substantial number of plant-specific TF families have been produced, and these TFs play a pivotal role in plant growth and development, as well as the stress response (Lehti-Shiu et al., 2017). The Trihelix TFs constitute a family of transcription factors that are predominantly found in plants. The Trihelix TF was initially identified in pea and was observed to the core sequence of 5′-G-Pu-(T/A)-A-(T/A)-3′ of the promoter region of the rbcS-3A gene which is a light-inducible and leaves-specific expression gene (Kaplan-Levy et al., 2012; Nagano, 2000). In addition, Trihelix TF was initially designated GT factor due to its ability to bind to photosensitive GT elements (Nagano, 2000). Researchers have revealed that the Trihelix structure of GT factors is highly similar to that of Myb/SANT-LIKE DNA-binding domains (Qin et al., 2014).

In recent years, the Trihelix TF has been identified and analyzed in a number of plant species, including Arabidopsis thaliana (Xu et al., 2018), Zea mays (Zhao et al., 2023), Oryza sativa (Kaplan-Levy et al., 2012), Sorghum bicolor (Li et al., 2021a), Chenopodium quinoa Willd (Li et al., 2022a) and Salix matsudana Koidz (Yang et al., 2023). Furthermore, the expression pattern of the Trihelix TF family in plant growth and development and various stress responses has been investigated, thereby providing valuable genetic resources for reverse genetics research (Qin et al., 2014). Furthermore, the Trihelix TF family is relatively conserved and can typically be divided into five subfamilies, GT1, GT2, GTγ, SH4, and SIP1 (Kaplan-Levy et al., 2012). It should be noted that the gene structure of the majority of Trihelix genes exhibits considerable variation between different plant species, particularly at the C-terminus (Li et al., 2021a). An understanding of the genetic diversity of TFs can facilitate the identification of valuable targets for the enhancement of crop traits, including yield, quality, and stress tolerance.

Studies have proved that Trihelix genes play an important role in plants. In Arabidopsis, ASIL1, a member of the Trihelix family, was observed to specifically recognize a GT element of the 2S albumin gene promoters, resulting in alterations to the globe gene expression profile (Xu et al., 2018). In addition, a trihelix transcription factor GT-2-LIKE1 (GTL1), and its homolog DF1 have been demonstrated to repress root hair growth by directly regulating the expression of gene the ROOT HAIR DEFECTIVE SIX-LIKE4. Loss-of-function mutants of GTL1 showed larger trichomes than the wild type (WT), which is associated with an increase in nuclear DNA content (Breuer et al., 2009). Similarly, Trihelix genes have also been demonstrated to play a role in the plant stress response. The GT1 subfamily genes in Arabidopsis are implicated in salt stress and pathogen responses (Murata, Takase & Hiratsuka, 2002). In rice, OsGTγ-2 functions as a positive regulator of salt stress responses, with a role in modulating this process by controlling the expression of ion transporters (Liu et al., 2020). Moreover, the GmGT-2A and GmGT-2B of soybean were found could contribute to stress tolerance by regulating of a set of genes (Xie et al., 2009). There were about 20 trihelix genes in Chrysanthemum that can respond to phytohormone treatments and abiotic stresses (Song et al., 2016). In maize, the ZmGT-3b, a GT factor, is associated with regulation of photosynthesis and defense response to Fusarium graminearum infection in maize seedling (Zhang et al., 2021). These studies show that the Trihelix genes possess a multitude of functions in plants. However, the precise molecular mechanism through which they operate remains unclear, necessitating further investigation.

Potatoes (Solanum tuberosum L.) are a Solanaceae crop that grows annually (Potato Genome Sequencing Consortium, 2011). Their tubers represent a crucial source of starch, protein, vitamins, and antioxidants and are also widely employed for vegetative propagation (Burlingame, Mouillé & Charrondière, 2009). However, the productivity of potatoes is significantly impacted by various abiotic stresses, such as drought and temperature extremes. These stresses can have a detrimental effect on potato growth, tuber development and overall yield (Nasir & Zoltan, 2022). Despite advancements in understanding plant stress responses, there remains a significant gap in identifying and harnessing specific molecular mechanisms that could enhance potato resilience to abiotic stresses. While various stress-responsive genes and pathways have been studied, the precise roles of specific TFs in regulating stress responses in potatoes are not fully elucidated (Chacón-Cerdas et al., 2020). In particular, the role of Trihelix TFs in potato stress responses is an underexplored area. A previous study have found that the promoter sequence of the Trihelix TF contains a large number of cis-acting elements associated with abiotic stress, hormonal cues, light exposure, and defensive mechanisms (Enghiad & Saidi, 2023). Furthermore, it is anticipated that certain Trihelix genes will exert a substantial influence on the abiotic stress tolerance of potato (Enghiad & Saidi, 2023). In addition, compared to the previous version, there has been a significant improvement in the quality of genome assembly and annotation for V6.1 potatoes genome. Therefore, we performed a comprehensive genomic analysis, leading to the identification of the Trihelix gene family. As a result, we identified 43 Trihelix genes in potato and systematically analyzed their phylogeny, gene structure and expression patterns in different tissues. Moreover, the transcriptional regulation network of Trihelix genes has been identified and analyzed. The findings of this study may serve as a foundation for subsequent functional analysis of potato development.

Materials and Methods

Identification of Trihelix genes in potato

The Hidden Markovmoder (HMM) profile (PF13837) of Myb/SANT-like DNA-binding domain was downloaded from InterPro (https://www.ebi.ac.uk/interpro/entry/pfam/PF13837). The protein sequences of potato (version 6.1) were downloaded from Spud DB (http://spuddb.uga.edu/). The protein sequences of potato were scanned using the HMM profile with the hmmsearch program of HMMER software. The sequences were selected as candidate Trihelix gene (StMSL) if their E-value was <0.00001. Further, the candidate sequences were further confirmed using SMART (Simple Modular Architecture Research Tool) and the InterPro database (Letunic, Khedkar & Bork, 2021).

Characteristics of the sequences and analysis of the conserved motif of StMSLs

The physiochemical properties of StMSLs protein sequences, including their length, molecular weight and theoretical isoelectric points (pIs), were analyzed using Expasy’s ProtParam server. The gene structure of StMSLs was analyzed by using the DNA sequences and their corresponding coding sequences with GSDS (Hu et al., 2015). To analyze the conserved domains of the StMSLs family members in potato, the MEME online tool was utilized, with a maximum of 10 motifs to identify (Bailey et al., 2015).

Phylogenetic analysis of the Trihelix family

The protein sequences of the Trihelix members from Arabidopsis, maize, rice, sorghum, grape, tomato and potato were aligned using ClustalX2 (Larkin et al., 2007). The resulting multiple sequence alignment was used for phylogenetic analysis. Phylogenetic analysis was conducted using the maximum likelihood method with FastTree2 software, and a bootstrap value of 1,000 was applied (Price, Dehal & Arkin, 2010). Then, the iTOL online tool was used to manage the phylogenetic tree (Letunic & Bork, 2021).

Chromosomal location and gene duplication of StMSLs

The genome annotation file for potato (DM_1-3_516_R44_potato.v6.1.hc_gene_models.gff3, V6.1) was downloaded from PGSC. Based on this genome annotation information, the chromosomal location of each StMSL gene was extracted. The chromosomal location was visualized using the MG2C online tool (Chao et al., 2021). For the gene duplication analysis, the genome and genome annotation file of potato (version 6.1) were used, and the Tbtools software was used to analyze the gene duplication event (Chen et al., 2023).

RNA-seq data analysis

The RNA-Seq data, which has been made publicly available through NCBI SRA database accession number SRA030516 was used to investigate the expression patterns of StMSLs in different potato tissues and osmotic stress (Potato Genome Sequencing Consortium, 2011). The aforementioned RNA-Seq data, which was quantified for each gene model in Transcripts Per Million (TPM), was downloaded from Spud DB (https://spuddb.uga.edu/). Prior to conducting the heatmap analysis, the TPM values of StMSLs were selected and transformed using log2(TPM+1). The transformed values were subsequently plotted in a graph using the TBtools software (Chen et al., 2023).

Construction of the Trihelix transcription factor regulatory network

The position frequency matrices (PFM) files of DNA binding preferences for Trihelix TF was searched and downloaded from the JASPAR database (Rauluseviciute et al., 2024). A Perl script was developed for the purpose of extracting sequences situated 2,000 bp upstream of the transcription start site, which were subsequently identified as promoter sequences.. The FIMO tool, which forms part of the MEME suite, employs prepared PFM files and promoter sequences to analyse the positions of motifs and to obtain information about Trihelix TF target genes, with a threshold of 10−5 (Bailey et al., 2015). The regulatory network of TF-genes based on gene expression data of StMSLs and the potential target gene from different potato tissues (callus, immature whole fruit, sepals, roots, tubers, carpels, inside of fruit, stolons, shoots, petioles, leaves, mature whole fruit, stamens, flowers and petals) was predicted using the GENIE3 R package with weight >0.01 (Huynh-Thu et al., 2010). The gene regulation network was visualized using Cytoscape (v 3.8.0) (Shannon et al., 2003). The Gene Ontology (GO) enrichment of the candidate target genes was identified utilising the AgriGO tool (Tian et al., 2017). The Fisher’s exact test was employed to ascertain the significance of the GO terms, with a P-value of less than 0.05.

Plant materials, stress treatments and qRT-PCR analysis

The sterile, tissue-cultured seedlings of the potato cultivar ‘Desiree’ were meticulously transplanted onto a perlite substrate following a 15-day cultivation period. The seedlings were initially nourished with the standard Hoagland nutrient solution for a subsequent 15-day period. The environmental conditions were meticulously controlled, with the temperature maintained at a constant 25 °C and a photoperiod of 16 h of light followed by 8 h of darkness. The seedlings that demonstrated consistent growth were selected and rinsed three times with a quarter-strength Hoagland nutrient solution. Subsequently, the seedlings were subjected to a 24-h treatment with 100 mM NaCl. To ensure the reliability of the findings, each experimental condition was replicated three times, with an equal volume of double-distilled water serving as the control. After the treatment, leaf tissues from each group were rapidly immersed in liquid nitrogen to halt enzymatic activity and preserve cellular integrity, before being stored at −80 °C for further analysis (Mei et al., 2021). Total RNA was extracted from the samples using Trizol Reagent (Invitrogen, Waltham, MA, USA). Following DNase I treatment, the integrity and quantity of the RNA were assessed using a NanoDrop 2000 spectrophotometer (Allsheng, Hangzhou, China). Approximately 4 µg of the isolated RNA were subjected to reverse transcription facilitated by M-MLV reverse transcriptase (Promega, Madison, WI, USA). Quantitative real-time polymerase chain reaction (qRT-PCR) was performed using SYBR Green (TaKaRa, Kusatsu, Japan) on a Bio-Rad CFX96 Touch Real-Time PCR Detection System (Bio-Rad, Hercules, CA, USA). The PCR protocol included an initial denaturation step at 95 °C for 10 min, followed by 40 cycles of denaturation at 95 °C for 15 s, annealing at 55 °C for 15 s, and extension at 72 °C for 30 s. Each sample was processed in triplicate to ensure repeatability. The relative expression levels of the genes were calculated using the 2−ΔΔCt method, with the potato Actin gene serving as an endogenous control for normalization. The primer sequences used for qRT-PCR in this study were listed in Table S1.

Results

Genome-wide identification of Trihelix TF family

To identify Trihelix TF members in potato, we used the hidden markov model of the Myb/SANT-LIKE domain (PF13837) as the query sequence in the HMMER program to search the potato whole protein sequences. We then confirmed the candidates using the SMART and InterPro to ensure they contained the Myb/SANT-LIKE domain. A total of 43 trihelix TF genes were identified and named StMSL1 to StMSL43 based on the position of the corresponding genes on chromosomes 1 to 12 (Tables 1, S2). The ExPASy server was used to calculate the physical parameters of each Trihelix protein. The full-length protein sequences of StMSL ranged from 106 (StMSL33) to 998 amino acids (StMSL13), and their molecular weights ranged from 11,497.1 to 111,520.6 Da. The pIs of the proteins varied greatly, ranging from 4.32 to 10.58 (Table 1).

Table 1 The physical and chemical properties of the members of the Trihelix family in potato.

Name	Gene ID	Protein length	MV (Da)	pI	
StMSL1	Soltu.DM.01G020620.1	308	34,721.2	10.13	
StMSL2	Soltu.DM.01G020660.1	335	37,603.3	10.21	
StMSL3	Soltu.DM.01G020710.1	253	28,928.4	9.16	
StMSL4	Soltu.DM.01G028200.1	683	76,530.7	9.96	
StMSL5	Soltu.DM.01G030180.1	329	37,097.1	9.45	
StMSL6	Soltu.DM.01G035650.1	344	37,805.1	9.92	
StMSL7	Soltu.DM.01G036430.1	339	38,941.3	7.85	
StMSL8	Soltu.DM.01G043960.1	392	45,199.6	7.36	
StMSL9	Soltu.DM.02G003400.1	338	38,311	8.86	
StMSL10	Soltu.DM.02G012410.1	341	38,812.5	8.38	
StMSL11	Soltu.DM.02G016000.1	283	32,699.8	5.57	
StMSL12	Soltu.DM.02G018190.1	497	56,274.3	10.58	
StMSL13	Soltu.DM.03G016380.1	998	111,520.6	7.4	
StMSL14	Soltu.DM.03G024640.1	852	94,364.3	8.34	
StMSL15	Soltu.DM.03G032480.1	349	39,631	5.27	
StMSL16	Soltu.DM.03G036820.1	373	41,269.8	9.89	
StMSL17	Soltu.DM.04G011000.1	327	37,551.3	4.71	
StMSL18	Soltu.DM.04G015330.1	316	35,443.2	6.28	
StMSL19	Soltu.DM.04G027080.1	628	69,388	6.38	
StMSL20	Soltu.DM.05G013300.1	352	40,568.1	4.42	
StMSL21	Soltu.DM.06G010060.1	273	31,288.9	9.59	
StMSL22	Soltu.DM.06G033750.1	326	37,859.7	6.28	
StMSL23	Soltu.DM.07G020890.1	397	44,403.1	10.36	
StMSL24	Soltu.DM.08G001080.1	414	46,317.4	9.51	
StMSL25	Soltu.DM.08G002920.1	134	15,585.6	10.34	
StMSL26	Soltu.DM.08G012280.1	472	55,191.2	6.32	
StMSL27	Soltu.DM.08G018560.1	158	17,998.7	4.32	
StMSL28	Soltu.DM.09G004040.1	542	62,435	6.93	
StMSL29	Soltu.DM.09G004470.1	533	57,982.5	6.67	
StMSL30	Soltu.DM.09G004490.1	496	55,952.8	6.72	
StMSL31	Soltu.DM.09G005630.1	519	58,369.9	7.18	
StMSL32	Soltu.DM.09G007650.1	298	34,972.3	7.72	
StMSL33	Soltu.DM.09G009310.1	106	11,497.1	10.53	
StMSL34	Soltu.DM.09G019790.1	282	32,977	5.31	
StMSL35	Soltu.DM.09G029350.1	440	51,867.5	6.28	
StMSL36	Soltu.DM.10G024830.1	415	48,785.2	8.15	
StMSL37	Soltu.DM.11G000160.1	389	44,073.2	4.85	
StMSL38	Soltu.DM.11G008900.1	503	57,413.2	8.06	
StMSL39	Soltu.DM.12G000660.1	443	50,001.6	6.01	
StMSL40	Soltu.DM.12G007690.1	649	72,215.4	6.67	
StMSL41	Soltu.DM.12G010580.1	374	43,010.7	4.64	
StMSL42	Soltu.DM.12G013620.1	178	20,229.7	8.83	
StMSL43	Soltu.DM.12G028480.1	359	39,476	9.64	

Conserved motifs and structural analysis of potato StMSLs

The MEME suite was utilized to identify conserved domains in the 43 StMSL protein sequences. A total of 10 conserved motifs were predicted, with lengths varying from 15 to 50 amino acids (Fig. 1). Motifs 1 and 2 were the most frequent in StMSL protein sequences, with motif I present in all genes and motif II present in all genes except StMSLS33. Additionally, the positions of motif 1 and motif 2 were relatively conserved. Furthermore, motifs 4 and 9 were only present in four StMSLs, and these two motifs were highly correlated, co-occurring in three genes (StMSL1, StMSL15 and StMSL18), suggesting a correlation between their functions. Additionally, a cluster analysis was conducted on the proteins of StMSL, revealing that they can be broadly classified into three categories. Category I can be further divided into three subclasses (Fig. 2). The genes that contain the motif 4 and 9 were found to be grouped together. Furthermore, the cDNA and genome sequences were used to analyze the structural diversity of StMSL genes. The analysis revealed that the genes had between one (StMSL1 and StMSL3) and 18 (StMSL14) exons. Notably, subfamily IA genes lacked 5′UTR, except for StMSL9 and StMSL27, while all IB genes contained 5′UTR. These findings suggest both motif conservation and gene structure diversity, which may be linked to their functional differentiation.

Figure 1 The conserved protein motifs in Solanum tuberosum L. Trihelix family (StMSLs).

Each motif is represented by a box in a different color, and the corresponding sequence is shown next to the box.

Figure 2 The neighbor joining phylogenetic tree and gene structure of StMSLs.

Phylogenetic relationships between the trihelix genes in potato and other plants

To understand the relationship between trihelix genes in different plant species, the protein sequences of the trihelixs from potato, Arabidopsis, tomato, grape, foxtail millet, sorghum and rice were analyzed using the maximum likelihood (bootstrap = 1,000). As shown in Fig. 3, almost all trihelix genes member of dicots and monocots could be separated in the evolutionary tree, indicating that trihelix family of dicots and monocots have distinct evolutionary differences. According to the topological tree structure and the classified marker of Arabidopsis, the trihelixs were clustered into six groups: SIP1, GT1, GT2, GTγ, SH4 and GT3. Among of these, SIP1 had the most number of the trihelixs (85), while GT1 had the fewest trihelixs members (23). In potato, the SIP1, GT1, GT2, GTγ and SH4 subfamilies contained 13, three, seven, seven, and 10 genes, respectively. It’s worth noting that the GT3 group which was not found in previous reports were only contained three StMSLs (StMSL4, StMSL12 and StMSL31) and one rice trihelix gene. These four genes contained one ortholog pair that formed by StMSL31/LOC_Os08g1700.

Figure 3 Phylogenetic tree of the Trihelix genes in potato, Arabidopsis, maize, rice, sorghum, grape, and tomato.

Chromosome distribution and gene duplication of Trihelix gene in potato

The chromosome positions of StMSL genes were studied using the genome annotation files of potato. The 43 StMSLs were found to be unevenly distributed on 12 chromosomes in potato. Chromosomes 1 (Chr01) and 9 (Chr09) contained the largest number of StMSLs (eight genes each), while Chr10 contained the least number of StMSLs (one gene) (Fig. S1). In addition, a pair of tandem repeats was present on chromosomes 1 and 9. Furthermore, five pairs of segmental duplication genes were identified which contained eight genes (StMSL9, StMSL10, StMSL11, StMSL19, StMSL23, StMSL38, StMSL40, and StMSL43) (Fig. 4). Further analysis of these gene subfamilies revealed that all of these genes are related to their respective subfamilies. Notably, StMSL19 belong to the GT2 subfamily and had two paralogs (StMSL38 and StMSL40), and StMSL23 which belong to the Sp1 subfamily also had two paralogs (StMSL23 and StMSL43).

Figure 4 Collinearity of duplicated gene pairs of StMSLs.

Gray lines show collinear gene pairs of potato, red lines show collinear gene pairs between StMSL.

Expression profiles of Trihelix genes in potato tissues

The RNA-Seq data publicly available was used to investigate the expression profiles of StMSLs in various potato tissues, including callus, tuber, root, sepal, fruit, shoot, stolon, leaves, petiole, stamen, flower, and petal. Out of the 43 StMSL genes, 36 expressed in at least one tissue (TPM > 1) (Fig. 5). The expressed genes could be divided into two groups, one group contains 18 genes with high overall expression, and the other group contains 18 genes with relatively low expression. In addition, we found that the tissues could also be divided into two clusters. The tissues related to leave and flower (leaves, petioles, stamens, flowers and petals) were clustered together. Moreover, the mature whole fruit clusters with shoots and stolons, but not with immature whole fruits or the interiors of fruits. Furthermore, we found that some of the StMSL genes exhibited a tissue-specific expression pattern and some were housekeeping genes that are expressed in all tissues. For example, StMSL6 was expressed in all the tissues studied, while StMSL30 was only highly expressed in the callus and tuber. StMSL12 showed high expressed in flower tissues (stamens and flowers), StMSL14 had higher expression in tissues with leaves (shoots and leaves), and StMSL10 was higher expression in fruit tissues (immature whole fruit, inside of fruit and mature whole fruit). Additionally, five StMSLs were highly expressed in the root. These results suggest that the diverse expression patterns of StMSLs may be associated with their role in the specific tissue development.

Figure 5 Expression pattern of StMSLs in 15 tissues of potato.

The red color indicates a gene with a high level of expression; the blue color indicates a gene with a low level of expression.

Generation and analysis of Trihelix transcriptional regulation network (TRN) in different tissues

To determine the TRNs underlying tissues of gene expression in potato, we generated tissues TRNs using available transcriptomics data. In total, 387 candidate target genes of 36 StMSL genes were included in our constructed TRN, with a total of 910 interactions (Fig. 6). To further investigate this observation, we performed gene function annotation of the target genes. We found that these genes were significantly enriched in a total of 44 GO terms with diverse functions (Fisher test P < 0.05) and the top 10 enriched GO terms as list in Table 2 (Figs. S2–S4). For instance, ten genes were found to be associated with the methylation of molecules. These genes included the genes Soltu.DM.10G012870, Soltu.DM.06.G015090 and Soltu.DM.03G014230, as well as the DNA methyltransferase gene Soltu.DM.08G003560. This suggests that these target genes play a crucial role in the molecular methylation process. Five additional genes are associated with regulating cell morphogenesis. For instance, the gene Soltu.DM.05G009310 was linked to steroid-mediated signalling pathways, and Soltu.DM.07G017700 is associated with pollen development. Furthermore, within the network, StMSL23 had the most interactions with 29 genes (Fig. 7). StMSL23 gene and the 29 candidate regulated genes were highly expressed in the immature whole fruit, indicating that StMSL23 may play a significant role in the immature whole fruit. These indicate that there is a complex regulatory network involving StMSLs.

Figure 6 Transcriptional regulatory network of StMSLs in potato.

Red triangles represent StMSLs; blue dots represent potato genes.

Table 2 The top 10 enriched gene ontology terms of candidate target genes.

GO term	Ontology	Description	
GO:0016701	F	Oxidoreductase activity, acting on single donors with incorporation of molecular oxygen	
GO:0016702	F	Oxidoreductase activity, acting on single donors with incorporation of molecular oxygen, incorporation of two atoms of oxygen	
GO:0004857	F	Enzyme inhibitor activity	
GO:0009738	P	Abscisic acid-activated signaling pathway	
GO:0098772	P	Molecular function regulator	
GO:0043414	C	Macromolecule methylation	
GO:0031123	P	RNA 3′-end processing	
GO:0005768	C	Endosome	
GO:0005773	C	Vacuole	
GO:0097306	P	Cellular response to alcohol	

Figure 7 Heatmap of the expression patterns of StMSL23 candidate regulatory genes.

Identification of trihelix genes involved in osmotic stress

To investigate the role of trihelix genes in the response to abiotic stress, we first analysed their expression patterns under osmotic stress including salt and mannitol using RNA-Seq data. The results indicated that 18 genes exhibited upregulation (fold change > 2) in response to at least one type of osmotic stress. This included nine genes that were upregulated in response to salt stress and 18 genes that were upregulated under mannitol stress conditions (Fig. 8A). For example, the StMSL18 and StMSL42 were up-regulated in both hyperosmotic stress. These findings suggest that trihelix genes may play a pivotal role in potato in response to hyperosmotic stress. Furthermore, to ascertain whether the expression of these genes can be induced under osmotic stress, qRT-PCR was employed to verify the expression patterns of eight genes under NaCl stress in potato shoot. The results showed that the expression of these genes, except StMSL43, were up-regulate to varying degrees under salt stress (Fig. 8B). Notably, StMSL4 and StMSL6 exhibited the most significant upregulation, more than 100-fold. In addition, we found that in the RNA-Seq data, the expression levels of MSL14 and MSL27 under high osmotic stress did not exceed twice than that in the control. However, through qRT-PCR analysis, we discovered that the expression of those were also upregulated under salt stress, indicating that there may be more trihelix genes involved in the potato’s response to salt stress.

Figure 8 The expression pattern of StMSLs under osmotic stress.

(A) RAN-Seq analysis of StMSLs in response to NaCl and mannitol; (B) qRT-PCR analysis of StMSLs in response to NaCl. *p < 0.05, **p < 0.01 (n = 3 independent replicates).

Discussion

Plants require dynamic and synergistic expression of intracellular genes to perform a wide variety of biological processes during growth. Transcription factors (TFs) play a crucial role in regulating gene expression (Spitz & Furlong, 2012). In plants, 59 plant transcription factor families have been identified (Jin et al., 2017). Some of these are plant-specific transcription factor families, such as the AP2 transcription factor family, the ERF transcription factor family, the WRKY transcription factor family and the Trihelix gene family. Among them, the Trihelix transcription factor family is widely distributed in plants and plays an important role in regulating plant growth and development as well as light response (Kaplan-Levy et al., 2012). However, the regulatory role of Trihelix TF in potato in biotic and abiotic stresses remains to be further elucidated (Chacón-Cerdas et al., 2020). Herein, bioinformatics analyses were employed to analyze the Trihelix gene family of potato at the whole-genome level.

In a previous study, a total of 22 trihelix gene family members were identified based on the potato genome of V4.03 (Enghiad & Saidi, 2023). The present study, which makes use of the latest assembled and annotated version of the potato genome (V6.1), has identified a total of 43 trihelix gene family members. This represents an increase on the 22 members of the trihelix gene family that were identified in previous studies. The identification of these new Trihelix gene family members provides a basis for further studies on the function of the trihelix gene in potato. To date, trihelix gene families have been identified in various plants, including the dicotyledonous species Arabidopsis thaliana (28) (Ali et al., 2016), Melilotus albus (34) (Zhai et al., 2023), and Brassica Rapa (52) (Wang et al., 2017), as well as the monocotyledonous species rice (31) (Fang et al., 2010) and corn (44) (Du, Huang & Liu, 2016). It was found that there is no positive correlation between the number of trihelix genes and genome size. This suggests that this gene family has undergone amplification or deletion events in some species during plant evolution. These results are consistent with our finding of two tandem duplication genes and five pairs of segmental duplication genes in potato. Further evolutionary analysis revealed that the trihelix genes can be divided into SIX groups: SIP1, GT1, GT2, GTγ, SH4 and GT3, with SIP1 being the largest subfamily and GT1 the smallest. Notably, we identified three previously less identified members of the GT3 family in potato, and this family member, StMSL31, could form a direct homologous pair with LOC_Os08g1700 in rice. This suggests that this class of genes may have been amplified from other genes during potato origin and may play a role in species formation.

Structural domain analysis of the potato trihelix gene family revealed that all these genes contain motifs 1 and 2, and the relative positions of these two motifs are conserved among the 40 trihelix genes, exactly opposite in StMSL13 and StMSL17. This suggests that these two structural domains may play a central role in the implementation of trihelix gene functions (Hu et al., 2023). In addition, some motifs were only present in some specific genes, for example motifs 4 and 9 were only present in four StMSLs and co-occurred in three genes (StMSL1, StMSL15 and StMSL18), suggesting that these two motifs may be functionally correlated in some way. This may represent a novel function that these genes have evolved to facilitate development or adaptation to the changing environment.

Studies have shown that trihelix TF can be involved in developmental processes and are key regulators of the stress response in plants, making them potential candidates for genetic improvement of crops (Li et al., 2022b; Nguyen et al., 2017; Zhang et al., 2021). For instance, it was demonstrated that GTL1, which belongs to the GT-2 subfamily in Arabidopsis, directly represses the expression of genes related to stomatal density and distribution, which are involved in the response to drought stress (Yoo et al., 2010). In maize, ZmThx20 encodes a GT-2 trihelix transcription factor. The absence of ZmThx20 results in abnormalities in endosperm development and the accumulation of storage reserves in seeds (Li et al., 2021b). However, the function of trihelix TF in potato is still unclear. We analyzed the expression pattern of trihelix genes in potato in 12 different tissues and found that StMSL12 was highly expressed in floral tissues (stamens and flowers); StMSL14 was highly expressed in leafy tissues (shoots and leaves). The homologous gene to StMSL14 in Arabidopsis is ATRNJ, which may predominantly regulate chloroplast gene expression at the post-transcriptional level through coordinated activity of nuclear-encoded ribonucleases and RNA-binding proteins (Halpert et al., 2019). StMSL10, which belongs to the GH4 subfamily, exhibited high expression levels in fruit tissues, including immature whole fruit, the interior of the fruit, and mature whole fruit. In the rice plant, SH4 is classified as a member of the SH4 subfamily. The recessive allele of SH4 has been observed to reduce the shattering of the mature seed head and the loss of ripe seeds prior to harvest (Li, Zhou & Sang, 2006). The identification of these tissue-specific expressed genes can provide useful genetic resources for the genetic improvement of potato. Additionally, our findings revealed that 18 genes exhibited a response to osmotic stress. Earlier research had demonstrated that certain trihelix TFs displayed responsiveness to a variety of abiotic stressors in potato (Enghiad & Saidi, 2023). Similarity, the transcription factor AtGT-3b has been demonstrated to play a role in the mediation of pathogen infection and salt stress in soybean and Arabidopsis through the regulation of the expression of the SCaM-4 gene (Park et al., 2004). However, the mechanism of how trihelix TF regulates potato development and response to adversity stress is still not clear. Therefore, identification of the downstream genes directly regulated by trihelix TFs is an urgent problem to be solved in order to elucidate the functions of such transcription factors in stress in potato. To gain further insight into the function of trihelix TFs in potato, we employed publicly available RNA-Seq data from different developmental stages of potato, with the aim of developing an initial understanding of the regulatory mechanisms of trihelix TFs in potato. By constructed the TRN of trihelix genes, we found that 387 genes are candidate target genes for 36 StMSL, and these genes form a large regulatory network. Functional annotation revealed a diversity of functions for these candidate genes, suggesting that they may play an important role in potato growth and development.

Conclusions

In conclusion, 43 trihelix genes were identified in potato. These genes had diverse motif distribution, and phylogenetic relationships that potentially contributed to their diverse functions. RNA-Seq data analyses revealed that some of the StMSLs were expressed in a tissue-specific pattern and thus were strongly related to tissue development. RNA-Seq and qRT-PCR results further indicating that StMSLs responded differently to osmotic stresses, including NaCl and mannitol treatment. This study provides baseline information for further studies on trihelix proteins in potato.

Supplemental Information

Supplemental Information 1 The chromosome location of Trihelix genes in potato.

Supplemental Information 2 The GO annotation of StMSLs candidate target genes at the biological process level.

Supplemental Information 3 The GO annotation of StMSLs candidate target genes at the molecular function level.

Supplemental Information 4 The GO annotation of StMSLs candidate target genes at the cellular component level.

Supplemental Information 5 The primers used in this study.

Supplemental Information 6 The location of StMSLs in the potato genome (V6.1).

Supplemental Information 7 MIQE checklist.

Additional Information and Declarations

Competing Interests

Author Contributions

Data Availability

The authors declare that they have no competing interests.

Chao Mei conceived and designed the experiments, performed the experiments, analyzed the data, prepared figures and/or tables, authored or reviewed drafts of the article, and approved the final draft.

Yuwei Liu conceived and designed the experiments, analyzed the data, prepared figures and/or tables, authored or reviewed drafts of the article, and approved the final draft.

Huiyang Song performed the experiments, analyzed the data, prepared figures and/or tables, and approved the final draft.

Jinghao Li performed the experiments, analyzed the data, prepared figures and/or tables, and approved the final draft.

Qianna Song performed the experiments, analyzed the data, prepared figures and/or tables, and approved the final draft.

Yonghong Duan performed the experiments, analyzed the data, prepared figures and/or tables, and approved the final draft.

Ruiyun Feng conceived and designed the experiments, authored or reviewed drafts of the article, and approved the final draft.

The following information was supplied regarding data availability:

The primers used in this study are available in the Supplemental File.

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
