# Peer review of "Genome-wide identification and expression analysis of the Trihelix transcription factor family in potato (Solanum tuberosum L.) during development"

_PeerJ, doi:10.7717/peerj.18578_

## Round 0.1 · original submission · Major Revisions

Please address concerns of the reviewers and amend manuscript accordingly.

·

Basic reporting

Ensure that the abstract effectively communicates the significance of the study and the relevance of Trihelix TF to breeding programs. In the introduction section please add some more points on the importance of TFs genetic diversity programs and the challenges associated with improving yield and abiotic stress response. Clearly articulate the gap in knowledge that the study aims to address and the specific objectives of the research. Highlight the problems associated with abiotic stress in potato and the reason why Trihelix TF can serve as a purpose to enhance development process. Try to streamline appropriate tables, figures, and statistical analyses to support the results and facilitate understanding. The research article should be improved by English native speaker to ensure a clear message to the audience.

Experimental design

Ensure that the methodology used for well-defined osmotic stresses such as salt and mannitol were clearly stated before citing the refence. Plant materials, treatments and samples collection should be in detail.

Validity of the findings

Critical consideration is needed to organize this section. You can provide a comprehensive discussion. You can provide a comprehensive discussion of the implications of the results in the context of Trihelix TF role in potato breeding programs. Analyze the significance of the Trihelix expression, its potential impact on osmotic stress response, and the prospects for developing improved potato varieties. Compare the findings with existing literature and discuss any discrepancies or novel insights. In conclusion, the section please try to summarize the main findings of the study and reiterate the importance of the identified StMSLs for empowering potato breeding programs. Clearly outline the implications for future research directions for functional analysis such as gene editing, transgenic breeding and the practical applications such as StMLS transformed potato tolerant to osmotic stress. Talking about references section ensures that the references cited in the manuscript are accurate, up-to-date, and relevant to the study.

Additional comments

Line 22-23: Rephrase
Line 40: Start the paragraph with a proper sentence “Benefit from advances in genomics research” is not proper
Line 54: Plant stress response instead of “stress response process of plants”.
Line : 55-56 rephrase the sentence .
Line 60: Factor not “factor”.
Line 106: Firstly delete.
Line 107: Then delete
Line 110: “Different potato tissues “please list the tissues used
Line 114: State the amount of mannitol and salt used for osmotic stress
Line 157: state the function.
Line 163 : classified instead of “classify”
Line 188: fruit and the inside of fruit. Wrong grammar
Line 216 : in response to at least one osmotic stress, wrong grammar
Line 229: TFs play an important role in the regulation of gene expression, wrong grammar.
Line 254-255: “It may 255 be a new function that these genes have evolved to development or adapt to the changing environment” Rephrase.

Reviewer 2 ·

Basic reporting

The manuscript describes comprehensive identification and characterization of the Trihelix transcription factors (TF) family in potato (Solanum tuberosum L.). By using public database, authors obtained the candidate genes that tissue-specifically expressed or responded to osmotic or salt stress. Subsequently, eight candidate StMSLs was validated by using qRT-PCR. I think that it is valuable in terms of providing a basis for understanding of the MSL gene family and the future scientific studies in potato. However, I have some concerns about the work, please see my comments below:
1. Identification of Trihelix genes in potato and their responses to abiotic stresses
has been published (Enghiad N, Saidi A. 2023. Analysis of Trihelix Genes and Their Expression in Potato in Response to Abiotic Stresses. Potato Research 66, 1075-1089.). The authors should discuss the differences between their work and the published paper, as well as highlight their own unique contributions.
2. L197 tissue-specific gene expression
3. L210 The conclusion -“three pairs of StMSLs genes can regulate each other” cannot be drawn only by bioinformatic analysis.
4. There are some typing, spacing, and language errors. Please carefully proof checks the entire text before resubmission. Such as L49, L50, L56, L57, L58, L59, L192, L199 should be Figure 6, L204, L206, L222, Figure 8 RNA-seq......

Experimental design

no comment

Validity of the findings

no comment

Reviewer 3 ·

Basic reporting

Urge the authors to clearly elucidate the knowledge gap and cite appropriate references to acknowledge the existing body of work.


Author's seem to suggest the knowledge gap in the abstract stating "the function of the Trihelix TF in potato (Solanum tuberosum L.) remains unknown".
There is existing literature elucidating function of trihelix genes in potato (see: Potato Research (2023) 66:1075–1089, https://doi.org/10.1007/s11540-023-09616-w).


Author's have identified 43 trihelix TFs in Potato, how do these compare to the 39 potato trihelix TFs reported in PlantTFDB (https://planttfdb.gao-lab.org/family.php?sp=Stu&fam=Trihelix).
A comparative analysis between the 43 trihelix TFs identified in the current analysis and the 39 existing Trihelix TFs reported in PlantTFDB would be useful for wider readership. In addition, kindly add commentary/systematic analysis or both on the benefits of using the prescribed methodology compared to the one used in PlantTFDB.

Experimental design

Request the authors to kindly explicate the design for RNAseq data analyses in the main manuscript.

The approach for TRN detection in unclear. Urge the authors to demonstrate the methodology adopted for the same.

Validity of the findings

Rewrite section "Generation and analysis of Trihelix transcriptional regulation network (TRN) in different tissues" with reference to appropriate figures. Also, GO analysis must be systematically supported by statistical validity. Kindly summarise the top hits output from GO analysis in a table.

Additional comments

Which cultivar of potato was used for validation with qRT-PCR?
Kindly use appropriate units for Fold Change in Fig. 8B. Use Volcano plot to depict
Units missing for heatmap colour bar in Fig. 5, Fig.7 and Fig. 8A.


Modify Abstract, Discussion, Methods and Conclusion accounting for the issues raised.

---

## Round 0.2 · accepted · Accept

All issues indicated by the reviewers were addressed and the manuscript is acceptable now.

Reviewer 3 ·

Basic reporting

'no comment'

Experimental design

'no comment'

Validity of the findings

'no comment'

Additional comments

Key issues related to the above sections have been addressed by the authors.
Minor typos and grammatical errors still persist. Encourage the authors to perform a rigorous proof-reading.